# Field Evaluation of Cotton Expressing Mpp51Aa2 as a Management Tool for Cotton Fleahoppers, *Pseudatomoscelis seriatus* (Reuter)

**DOI:** 10.3390/toxins15110644

**Published:** 2023-11-05

**Authors:** Brady P. Arthur, Charles P. Suh, Benjamin M. McKnight, Megha N. Parajulee, Fei Yang, David L. Kerns

**Affiliations:** 1Department of Entomology, Texas A&M University, College Station, TX 77843, USA; bparthur@tamu.edu; 2USDA-ARS Southern Plains Agricultural Research Center, College Station, TX 77845, USA; charles.suh@usda.gov; 3Department of Soil and Crop Sciences, Texas A&M University, College Station, TX 77843, USA; benjamin.mcknight@ag.tamu.edu; 4AgriLife Research and Extension Center, Texas A&M University, Lubbock, TX 79403, USA; megha.parajulee@ag.tamu.edu; 5Department of Entomology, University of Minnesota, St. Paul, MN 55108, USA; yang8905@umn.edu

**Keywords:** Mpp51Aa2, ThryvOn, cotton fleahopper, *Psuedatomoscelis seriatus*, *Gossypium hirsutum*

## Abstract

The cotton fleahopper (*Pseudatomoscelis seriatus* Reuter) is considered a highly economically damaging pest of cotton (*Gossypium hirsutum* L.) in Texas and Oklahoma. Current control methods rely heavily on the use of foliar-applied chemical insecticides, but considering the cost of insecticides and the critical timeliness of applications, chemical control methods are often not optimized to reduce potential yield losses from this pest. The *Bacillus thuringiensis* (*Bt*) Mpp51Aa2 (formerly Cry51Aa2.834_16) protein has proven effective against thrips and plant bugs with piercing and sucking feeding behaviors, but the impact of this toxin on cotton fleahoppers has not been investigated. To evaluate the Mpp51Aa2 trait effectiveness towards the cotton fleahopper, field trials were conducted in 2019, 2020, and 2021, comparing a cotton cultivar containing the Mpp51Aa2 trait to a non-traited isoline cultivar under insecticide-treated and untreated conditions. Populations of cotton fleahopper nymphs and adults were estimated weekly by visually inspecting cotton terminals. Square retention was also assessed during the first week of bloom to provide some insight on how the *Bt* trait may influence yield. While cotton fleahopper population differences between the traited and non-traited plants were not consistently noted during the pre-bloom squaring period, there was a consistent increase in square retention in cotton expressing Mpp51Aa2 relative to non-traited cotton. Additionally, cotton expressing Mpp51Aa2 offered similar square protection relative to non-traited cotton treated with insecticides for the cotton fleahopper. These findings indicate that the Mpp51Aa2 protein should provide benefits of delayed nymphal growth, population suppression, and increased square retention.

## 1. Introduction

For decades, the cotton fleahopper (*Pseudatomoscelis seriatus* Reuter) has been considered one of the top five most damaging insect pests of cotton in Texas, resulting in an average reduction of 100,577 bales annually from 2010 to 2021 [1,2,3,4,5]. In 2021 alone, this pest reportedly infested 98% of cotton acres across the state, resulting in a loss of 494,000 bales of cotton. Estimated crop losses from this reduction in bale production was equivalent to USD 266.5 million, 61% of the total lost cotton revenue in Texas [4]. While some efforts have shown promise in selecting for natural resistance mechanisms, the primary strategy for managing this persistent pest has been through insecticidal control [6,7,8]. However, the average cost of foliar treatments for this pest has risen from USD 4.72 per acre in 2010 to USD 15.19 per acre in 2021 [4,5]. Consequently, there has been a growing interest in developing a transgenic trait to control the cotton fleahopper [9].

The cotton fleahopper is a generalist feeder with a host range of over 160 species of plants. The adults move from one species of host to another, based on the time of year [10]. Despite the wide host range, cotton fleahoppers have preferred host species, including woolly croton (*Croton capitatus* Michx), horsemint (*Monarda punctata* L.), parthenium ragweed (*Parthenium hysterophorus* L.), silverleaf nightshade (*Solanum elaeagnifolium* Cav.) and others. The cotton fleahopper is initially observed and monitored in cotton during the early portion of the cotton growing season, when their preferred wild host species are less abundant [11,12,13]. Cotton fleahoppers generally feed on the newly formed pre-floral structures known as squares, as well as the apical meristem. Feeding on the squares can cause abscission of squares and ultimately the loss of the fruit, while feeding on the plant terminal can shorten internodes, cause abnormal node formation, and potentially delay maturity [11]. Like most Hemipteran: Miridae pests, the cotton fleahopper feeds using a piercing and sucking stylet which allows penetration of the plant tissues and ingestion of the sap [10]. After stylet insertion, the cotton fleahopper injects saliva into the plant tissues. This saliva contains enzymes that digest the plant cell walls and aid in initiating the flow of plant fluids [14,15].

Since 1996, the use of plant-incorporated insecticidal proteins derived from soil bacteria *Bacillus thuringiensis* (Berliner) (*Bt*) has provided excellent control for insect pest species in cotton and corn, reducing the necessity for foliar applications of insecticides [16]. However, these *Bt* traits have exclusively been introduced to target coleopteran and lepidopteran species with no activity on hemipterans [17]. With Bayer CropScience’s (St. Louis, MO, USA) discovery of the *Bt* protein Mpp51Aa2 (formerly referred to as Cry51Aa2) [18] and its variants, there is potential to utilize a plant-incorporated protein to control hemipteran and thysanopteran pests in cotton [9,19]. Similar to other Cry proteins, Mpp51Aa2 protein is a β pore-forming toxin acting on the epithelium of the insect midgut [17]. Utilizing an artificial diet containing the Mpp51Aa2.834_16 variant of the protein, Gowda et al. [17] determined that the lethal concentrations of the toxin towards the western tarnished plant bug, *Lygus hesperus* (Knight) and the tarnished plant bug, *Lygus lineolaris* (Palisor de Beauvois), were LC_50_ = 0.30 µg mg^−1^ and LC_50_ = 0.853 µg mg^−1^, respectively. In another study using an artificial diet, Graham et al. [20] showed that the presence of Mpp51Aa2 protein did not affect the feeding of first- or third-instar tarnished plant bug nymphs. However, the adults preferred a diet that did not contain the Mpp51Aa2 protein. This preference for non-*Bt* was further confirmed by Graham et al. [20], in observations that tarnished plant bug adults favored non-*Bt* excised squares and laid significantly more eggs in non-*Bt* diet packs. In field trials, the Mpp51Aa2 cotton variant, termed as ThryvOn (Bayer CropScience, St. Louis, MO, USA), has been shown to reduce the population density of tarnished plant bug adults by 23% relative to non-traited cotton [21]. Similarly, a 19-fold reduction in tarnished plant bug large nymphs (fourth and fifth instar) was observed in Mpp51Aa2 cotton relative to a non-traited cultivar [22]. Furthermore, cotton that contained the Mpp51Aa2 trait had higher levels of square retention relative to a non-traited isoline when exposed to tarnished plant bugs. When integrated into a pest management system using foliar insecticides, treatments containing the Mpp51Aa2 trait required 50% fewer insecticide applications relative to their non-traited counterparts [23]. Because the cotton fleahopper shares similarities in biology and feeding strategies with the tarnished plant bug and the western tarnished plant bug [24], it is plausible that cotton expressing the Mpp51Aa2 protein may exhibit similar activity against the cotton fleahopper. In a recent study by Bachman et al. [22], they did not observe any differences in the number of large cotton fleahopper nymphs between Mpp51Aa2 and non-traited cotton, but noted that Mpp51Aa2 had a 1.7-fold decrease in adult cotton fleahoppers in the subsequent generation. The objective of this study was to determine the efficacy of Mpp51Aa2-traited cotton on cotton fleahopper populations and explore the potential integration of this new *Bt* technology into a comprehensive pest management system.

## 2. Results

### 2.1. Cotton Fleahopper Populations

In 2019, the percentage of infested terminals was not significant for any life stage of cotton fleahopper during the first week of squaring (Table 1). The insecticide application following the first week of squaring significantly reduced populations of both small nymphs (*p* = 0.042) and large nymphs (*p* = 0.026), leading to differences in total cotton fleahoppers (*p* = 0.008) during the second week of squaring. During the third week of squaring, a significant interaction between the spray and trait effects was noted for small nymphs (*p* = 0.024) and large nymphs (*p* = 0.011), while only a spray effect was evident for adults (*p* = 0.001) and the total cotton fleahopper population (*p <* 0.001). When examining populations in the fourth week of squaring, a significant trait effect was observed for the number of large nymphs (*p* = 0.013). Additionally, a significant interaction between the spray treatment and trait was noted for adults (*p <* 0.001) and total cotton fleahoppers (*p <* 0.001), with the non-traited untreated treatment combination revealing significantly higher cotton fleahopper densities compared to all other treatment combinations.

In 2020, there was a significant trait effect on the number of adults (*p* = 0.004) and total cotton fleahoppers (*p* = 0.009) during the first week of squaring (Table 2). Similarly, during the second week of squaring, significant trait effects were observed for the number of adults (*p* = 0.031) and total cotton fleahoppers (*p* = 0.025). Following the first insecticide application during the second week of squaring, the insecticide treatment significantly reduced the numbers of large nymphs (*p* = 0.016), adults (*p* = 0.001), and total cotton fleahoppers (*p* = 0.003) during the third week of squaring. Cotton fleahopper counts for the fourth week of squaring showed that there was a significant spray effect on populations of small nymphs (*p <* 0.001), large nymphs (*p <* 0.001), and the total cotton fleahoppers (*p <* 0.001). There was also a significant interaction of the spray and trait effects on adults (*p* = 0.040). However, there were no significant treatment effects on the number of adults during the fifth week of squaring, although a significant spray treatment effect was observed for small nymphs (*p* = 0.001), large nymphs (*p <* 0.001), and total cotton fleahoppers (*p <* 0.001) during that week.

In 2021, significant trait effects were initially observed during the first week of squaring for the number of small nymphs (*p* = 0.024), adults (*p* = 0.005), and total cotton fleahoppers (*p* = 0.004) (Table 3). The first insecticide application made during the first week of squaring significantly reduced populations of all cotton fleahopper life stages during the second week of squaring; small nymphs (*p* = 0.001), large nymphs (*p* = 0.005), adults (*p* = 0.002), and total cotton fleahoppers (*p* = 0.001). During the third week of squaring, significant insecticide effects were observed on the abundance of small nymphs (*p <* 0.0001), large nymphs (*p* = 0.012), and total cotton fleahoppers (*p* = 0.001), but no significant effect was detected for adults (*p* = 0.349). In the fourth week of squaring, a significant interaction between trait and insecticide treatments was evident for all life stages; small nymphs (*p* = 0.034), large nymphs (*p* = 0.005), adults (*p* = 0.044), and total cotton fleahoppers (*p* = 0.004). Total cotton fleahoppers in the non-traited untreated treatment significantly exceeded that in all other treatment combinations.

Across all three years and sample dates for the untreated treatments, the ratio of small nymphs (≤third instar) to large nymphs (fourth and fifth instar) was 2.6:1 (small nymphs:large nymphs) in the ThryvOn treatment and 1.1:1 in the non-traited treatment (Figure 1). A significant chi-square analysis (χ^2^ = 15.45, df = 1, *p <* 0.0001) revealed the ThryvOn cultivar had a significantly lower proportion of large nymphs relative to small nymphs compared to the non-traited cultivar.

As there were significant insecticide effects on cotton fleahopper populations, our analysis of cumulative insect days (CIDs) considered only the untreated treatments. A one-way ANOVA, followed by post hoc Student’s *t*-tests, was used to determine differences in CIDs between ThryvOn and the non-traited for each of the life stages (small nymphs, large nymphs, and adults). There were no significant differences in CIDs between the non-traited and ThryvOn for small nymphs (*p* = 0.109) or across all cotton fleahopper life stages (*p* = 0.081) (Figure 2A,D). However, CIDs were significantly reduced for large nymphs (*p* = 0.026) and adults (*p* = 0.017) in ThryvOn compared to the non-traited cultivar (Figure 2B,C).

### 2.2. Square Retention

Square retention values estimated at first bloom consistently showed the non-traited, untreated treatment had the lowest square retention across all three years (Table 4). In 2019, there were significant trait (*p <* 0.0001) and insecticide treatment (*p <* 0.0001) effects, but no significant interaction (*p* = 0.898). Square retention for the ThryvOn cultivar was significantly higher at 66% compared to 46% for the non-traited (*p <* 0.0001). In 2020, there was no significant trait effect (*p* = 0.266), but significant insecticide (*p* = 0.020) and interaction (*p* = 0.047) effects were detected. Closer examination of the significant interaction revealed that square retention of the non-traited, untreated treatment (69%) was significantly reduced in comparison to the non-traited, treated treatment (86%). Similarly, in 2021, there was no significant trait effect (*p* = 0.215), but both insecticide treatment (*p* = 0.002) and interaction (*p* = 0.013) effects were significant. Square retention in the non-traited, untreated treatment (87.63%) was significantly lower than those observed for the non-traited treated, ThryvOn treated and ThryvOn untreated treatments. The ThryvOn untreated treatment was statistically similar to the non-traited treated and ThryvOn treated treatments.

Regression analyses of the cumulative insect days (Figure 2D) against square retention in the untreated treatments produced significant relationships (Figure 3). The regression model for ThryvOn (R^2^ = 0.8732) was (SR = 66.53 × e^−0.00051*CID^ + 24.52), where SR represents square retention and CID indicates cumulative insect days. The model for the non-traited (R^2^ = 0.9300) was (SR = 70.01 × e^−0.001*CID^ + 25.57). The two models were significantly different, based on the analysis of the extra-sum-of-squares F-test (*F =* 3.769, df = 3, 26, *p* = 0.022). The difference in the span (difference between the Y intercept and the plateau) of the two models suggests the ThryvOn treatment would shed fewer squares compared to non-traited treatments as the number of cumulative insect days increases. The half-life of the ThryvOn and non-traited models was 1366 and 669.7 CIDs, respectively, suggesting that ThryvOn requires a higher number of cumulative insect days to inflict a similar magnitude of square abscission.

## 3. Discussion

The population counts for 2019, 2020, and 2021 showed varying levels of infestation, characterized as relatively high in 2019, moderate in 2020, and relatively low in 2021. As the season progressed in each year of the study, differences in nymphal and adult population levels in the non-sprayed plots became more evident between the ThryvOn and non-traited (Table 1, Table 2 and Table 3). The high nymphal and adult populations observed during the squaring period in 2019 indicated that ThryvOn alone does not prevent cotton fleahopper colonization. Nevertheless, there is a notable benefit even in high populations when ThryvOn is coupled with insecticides, as cotton fleahopper populations increased at a significantly slower rate compared with those in the non-traited, treated cultivar, thus limiting the number of large nymphs and adults at the end of the squaring period.

Compared to 2019, infestations in 2020 were moderate, but still reached threshold levels on a consistent basis. Early in 2020, before cotton fleahopper populations reached spray treatment thresholds, there was an apparent trait effect on the number of adults. Interestingly, adult populations in ThryvOn treatments were significantly higher than those observed in the non-traited treatments. Possible explanations for this unexpected observation may be related to initial dispersal behaviors of cotton fleahoppers into the cotton field from the weeds along the field margins. Parajulee et al. [25] indicated that adult populations of cotton fleahoppers can exhibit a clumped distribution during the initial stages of infestation but that adults tend to disperse throughout the field and exhibit a more uniform distribution as the season progresses. Nevertheless, following the initial application of insecticide and as the squaring season progressed, there was a significant interaction of trait and spray effects on adult populations, where the non-traited, untreated treatment had a 2.2-fold increase in adults compared to the ThryvOn, untreated treatment.

The low levels of infestation throughout the 2021 growing season can be attributed to the diverse non-cotton vegetation in field margins. Ample rainfall during the growing season not only suppressed populations by washing smaller nymphs from the terminal [26], but also potentially increased the diversity and intensity of weedy species adjacent to the cotton field, particularly horsemint and woolly croton, not only serving as a source of cotton fleahoppers but also acting as a trap crop [27]. Although populations of cotton fleahoppers in our test plots were considerably lower in 2021 compared to the previous two years, as the season progressed, dispersal into the field indicated a significant interaction of the trait and spray effects, resulting in higher populations in the non-traited, untreated. In our study, the non-traited, untreated treatment possessed the highest numbers of cotton fleahoppers by the end of the squaring period in each year of the study. This is similar to the results found by Gowda et al. [17], where lower numbers of tarnished plant bugs and western tarnished plant bugs of all life stages were observed in ThryvOn cotton compared to a non-traited control. Likewise, Graham and Stewart [21] and Graham et al. [20] found that ThryvOn had fewer tarnished plant bug adults than the non-traited control, suggesting that the trait had a repellency effect on the pest, with adults preferentially feeding on the non-traited plants. This same result was reported in cotton fleahoppers, where the adult populations in ThryvOn were significantly lower than in the non-traited [28]. In contrast, the adult cotton fleahopper populations at the end of the squaring period in each year of our study suggest the singular *Bt* trait effect did not significantly suppress adult cotton fleahopper populations. Although no differences were observed in adult cotton fleahoppers between the traited and non-traited treatments, a significant interaction occurred between spray treatment and the *Bt* trait. This interaction suggests that there is a benefit to the ThryvOn trait in slowing the development of infestations. This observation aligns with Graham and Stewart [21], who reported a delay in cotton fleahopper reinfestation when ThryvOn cotton was over-sprayed with insecticide, potentially reducing the number of insecticide applications required to manage populations. Similar to the results of Asiimwe et al. [28] populations of nymphs, small or large, did not consistently show significant differences based on trait. However, differences were observed in CIDs for both large nymphs and adults in this study. It has been documented that the presence of *Bt* in the host plant can delay insect development [29], which suggests that the ThryvOn trait was largely responsible for the delayed nymphal development. In ThryvOn, CIDs for large nymphs and adults were reduced by 160 and 111 days, respectively, compared to those estimated for large nymphs and adults in the non-traited. These differences potentially contributed to the increase in square retention in ThryvOn, as large-nymph and adult stages of mirids are known to cause the highest magnitude of damage [22,30]. Given that the ratio of small to large nymphs was 2.6:1 in the ThryvOn treatment but closer to 1:1 in the non-traited cultivar, the rate of nymphal growth appeared to be more rapid and consistent in the non-traited plots. These findings align with those of Whitfield [31], who reported different ratios, small vs. large, of tarnished plant bug nymphs in ThryvOn compared to non-traited. Similarly, Jerga et al. [29] showed that the presence of Mpp51Aa2 protein delayed the development of early-instar tarnished plant bug nymphs.

A potential concern with the results is the presence of other insect pests; however, populations of tarnished plant bugs and western tarnished plant bugs were not observed in the test area for any of the years the field trials were conducted. The presence of late-instar cotton fleahopper nymphs in ThryvOn could potentially point to concerns with the longevity of the technology. As documented for other pest species, low doses of the toxins only eliminate the highly sensitive insects, resulting in potential evolution of resistance [32]. Killing only the homozygous susceptible populations and not the heterozygous resistant will allow for a faster increase in the resistant allele frequency within the pest population [33]. Failure to meet the requirements of a high-dose resistance management strategy is one of the main factors in field-evolved resistance to *Bt* toxins [34,35].

## 4. Conclusions

In summary, the incorporation of a *Bt* protein into cotton, with activity against sap-feeding insects like the cotton fleahopper, addresses a long-standing need in pest management. The Mpp51Aa2 protein provides benefits for the control of mirid pests of cotton, including delayed field colonization, nymphal development, population reduction, decreased dependance on foliar insecticides, and mitigation of insect-induced fruit abscission. The results of this study confirm that the ThryvOn trait suppresses populations of cotton fleahoppers, which is consistent with the suppression observed in previous work with other piercing–sucking insect pest species. Implementation of this new *Bt* technology, as part of an integrated pest management system for cotton fleahoppers, will potentially reduce the number of insecticide applications, provide producers with more flexibility to make timely applications of insecticides, and limit potential yield losses associated with cotton fleahopper feeding damage.

## 5. Materials and Methods

### 5.1. In-Season Data Collection

A three-year (2019–2021) field study was conducted in the Brazos Valley, Texas, to determine if Mpp51Aa2-traited cotton has an impact on cotton fleahopper populations and square retention. The experiment utilized near-isolines of Mpp51Aa2-traited and non-traited cotton with or without insecticidal control of the cotton fleahopper. The Mpp51Aa2-traited cotton cultivars, hereafter referred to as ThryvOn, used in the study were 18R445B3XF, 19R326LB3XF, and DeltaPine 2131B3TXF in 2019, 2020 and 2021, respectively. The non-traited cotton cultivars included an unspecified isoline in 2019, and the near-isoline, DeltaPine 2055B3XF, in 2020 and 2021. All seeds were sourced from Bayer CropScience, had been treated with fungicides and insecticides to prevent early-season pathogens and pre-fruiting pests (Acceleron Standard: Bayer CropScience, St. Louis, MO, USA), and contained the lepidopteran active *Bt* traits Cry1Ac, Cry2Ab2, and Vip3a19. These *Bt* proteins expressed in both cultivars are presumed to have negligible influence on our results, as it is documented that Cry1Ac, Cry2Ab and Vip3Aa are active on lepidopteran pests but not active on mirids such as cotton fleahoppers [19]. The experiment employed a randomized complete block design with all combinations of the two main effects of trait, ThryvOn and non-traited, and an insecticide regimen, treated and untreated. Treatment combinations included ThryvOn treated, non-traited treated, ThryvOn untreated, and non-traited untreated with five, three, and eight replicates in 2019, 2020, and 2021, respectively. Insecticidal treatments were applied when cotton fleahopper populations of the treated plots reached or exceeded 5 percent infested terminals; however, precipitation and equipment failure prevented timely application in some instances. Treatments designated as ‘treated’ were sprayed using a ground-driven high-clearance sprayer with commercially available insecticides that had proven effectiveness in cotton fleahopper management, as indicated by prior research conducted at Texas A&M AgriLife Extension [36]. Foliar insecticide treatments were a tank mix of 44.7 g ai per ha of acephate (Orthene 97, AMVAC Chemical Corporation, Newport Beach, CA, USA) and 60.5 g ai per ha of imidacloprid (Admire Pro, Bayer Crop Science, Raleigh, NC, USA). Each experimental plot was approximately 0.2 ha with a row spacing of 1.02 m. The seeding rate was approximately 13 seeds per meter. Applications of fertilizers, herbicides, growth regulators, and insecticides to manage cotton, weeds and other insect pests followed the recommendations of Texas A&M AgriLife Extension [36,37,38].

Cotton terminal inspections for cotton fleahoppers began at first week of squaring and continued on a weekly basis until first bloom. In 2019, 25 terminals were inspected per treatment plot. In 2020 and 2021, 100 terminals were inspected in each plot to reduce data variability. The total numbers of cotton fleahopper adults, large nymphs (fourth and fifth instar) and small nymphs (≤third instar) in each plot were recorded on each sample date. Cotton fleahopper instars were determined by the size of the head capsule and relative development of wing pads; first, second, and third instars have a narrower head and no presence of wing pads; fourth-instar nymphs have short wing pads that only extend to the base of the second abdominal segment, whereas fifth-instar nymphs have a head equal or greater in width relative to the abdomen, with long wingpads extending to at least the fourth abdominal segment [39,40]. All cotton fleahopper counts for each plot were converted to percent infested terminals by dividing the number of cotton fleahoppers encountered by the number of terminals inspected. Cumulative insect days (CIDs) were calculated using the method described by Ruppel [41], to highlight differences in overall insect pressure across different sample dates.

During the fourth week of squaring, percent fruit retention was estimated by dividing the number of squares present by the total number of fruiting positions on whole plants, with a specific focus on sympodial branches (fruiting branches) [42]. In 2019, five random plants were sampled per plot, whereas twenty random plants per plot were sampled in 2020 and 2021, to reduce variability in the data.

### 5.2. Statistical Analyses

All statistical analyses were performed utilizing GraphPad Prism 9.3.0 [43]. The percent cotton fleahopper infested terminals and percent fruit retention data were analyzed using a two-way ANOVA with trait, spray regimen, and their interaction as fixed effects. When comparing cumulative insect days by life stage, the main effects of trait, insecticide treatment, and the interaction of trait and insecticide treatment were designated as fixed effects, while year was designated as a random effect. Significant differences between treatment means were separated using Tukey’s honest significant difference method (α = 0.05). A chi-square test of independence (α = 0.05) was performed to examine differences in the small- and large-nymph population dynamics on ThryvOn and non-traited. Regression curves, generated by plotting square retention against cumulative insect days, were compared using an extra sum-of-squares F-test with an α = 0.05. The extra-sum-of-squares F-test compared the two regression curves to a simplified regression curve of the entire data set.

## Figures and Tables

**Figure 1 toxins-15-00644-f001:**
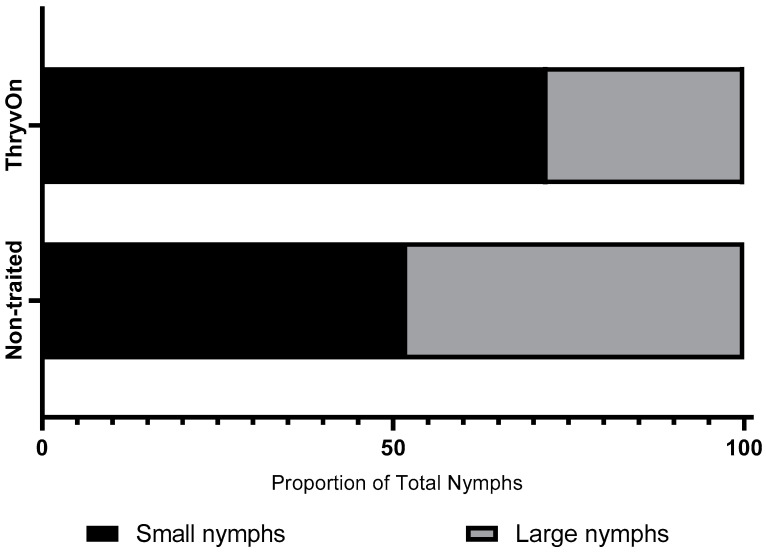
Chi-square analysis of the proportion of small nymphs (≤3rd instar) and large nymphs (4th and 5th instars) between non-traited untreated and ThryvOn untreated treatments across all years and sample dates (*n* = 376, χ^2^ = 15.45, *p* < 0.0001).

**Figure 2 toxins-15-00644-f002:**
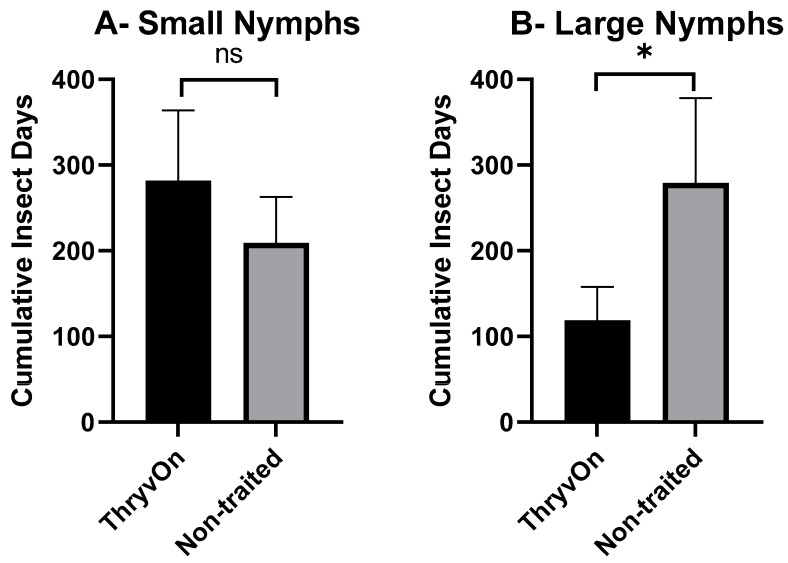
Cumulative insect days (mean ± SEM) of non-insecticide treated plots across all three years by cotton fleahopper life stage. (**A**) Small nymphs ≤ 3rd instar. (**B**) Large nymphs 4th and 5th instar. (**C**) Adult cotton fleahoppers. (**D**) All life stages combined. Significant differences between traits are denoted by “*” (ANOVA, Student’s *t*-test, *p* < 0.05).

**Figure 3 toxins-15-00644-f003:**
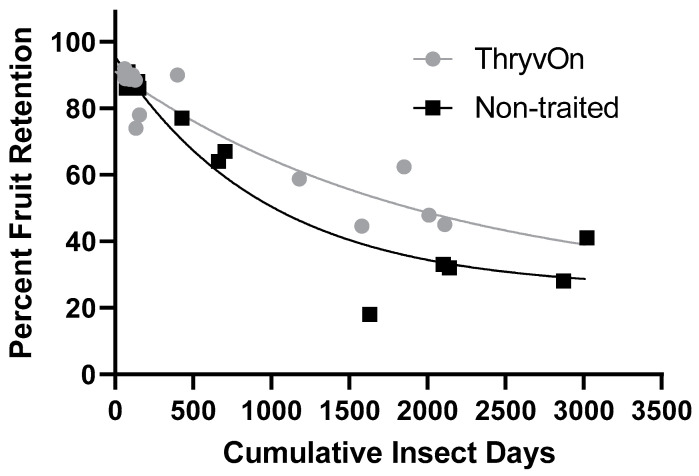
Relationship of cumulative insect days of cotton fleahoppers, across life stages and years, and fruit retention on untreated ThryvOn and non-traited cotton cultivars. An extra-sum-of-squares F-test was used to determine that the single-phase decay model shown that best fits each of the cultivars were significantly different (*p* = 0.02). ThryvOn (y = 66.53 × e^−0.00051*x^ + 24.52) (R^2^ = 0.8732), non-traited (y = 70.01 × e^−0.001*x^ + 25.57) (R^2^ = 0.9300).

**Table 1 toxins-15-00644-t001:** Percent (mean ± SEM) cotton fleahopper infested terminals by insect stage and week of squaring during 2019.

Treatment	First Week of Squaring			
	Small nymphs	Large nymphs	Adults	Total
Non-traited Treated	52.00 ± 13.56	24.00 ± 7.48	24.00 ± 11.66	100.00 ± 20.98
Non-traited Untreated	40.00 ± 6.32	12.00 ± 4.90	40.00 ± 8.94	92.00 ± 17.44
ThryvOn Treated	44.00 ± 7.48	12.00 ± 8.00	24.00 ± 14.70	80.00 ± 14.14
ThryvOn Untreated	36.00 ± 11.66	8.00 ± 4.90	16.00 ± 7.48	60.00 ± 17.89
Trait *p value*	0.565	0.235	0.293	0.163
Insecticide *p value*	0.341	0.235	0.722	0.443
Trait * Insecticide *p value*	0.857	0.546	0.293	0.740
	Second week of squaring			
	Small nymphs **	Large nymphs **	Adults	Total **
Non-traited Treated	4.00 ± 4.00	0.00 ± 0.00	36.00 ± 11.66	40.00 ± 0.00
Non-traited Untreated	16.00 ± 7.48	32.00 ± 13.56	52.00 ± 18.55	100.00 ± 13.56
ThryvOn Treated	12.00 ± 4.90	4.00 ± 4.00	16.00 ± 7.48	32.00 ± 4.00
ThryvOn Untreated	28.00 ± 8.00	12.00 ± 8.00	48.00 ± 14.97	88.00 ± 8.00
Trait *p value*	0.133	0.339	0.397	0.611
Insecticide *p value*	0.042	0.026	0.101	0.008
Trait * Insecticide *p value*	0.756	0.159	0.570	0.919
	Third week of squaring			
	Small nymphs ***	Large nymphs ***	Adults **	Total **
Non-traited Treated	8.00 ± 4.90 bc	0.00 ± 0.00 c	8.00 ± 8.00	16.00 ± 7.48
Non-traited Untreated	32.00 ± 4.90 ab	64.00 ± 9.80 a	44.00 ± 7.48	140.00 ± 17.89
ThryvOn Treated	0.00 ± 0.00 c	0.00 ± 0.00 c	4.00 ± 4.00	4.00 ± 4.00
ThryvOn Untreated	60.00 ± 12.65 a	28.00 ± 8.00 b	36.00 ± 11.66	124.00 ± 18.33
Trait *p value*	0.185	0.011	0.477	0.315
Insecticide *p value*	0.0001	0.0001	0.001	0.0001
Trait * Insecticide *p value*	0.024	0.011	0.811	0.884
	Fourth week of squaring			
	Small nymphs ***	Large nymphs *	Adults ***	Total ***
Non-traited Treated	12.00 ± 4.90 a	16.00 ± 4.00	44.00 ± 14.7 b	72.00 ± 18.55 b
Non-traited Untreated	40.00 ± 10.95 a	56.00 ± 17.20	140.00 ± 14.14 a	236.00 ± 32.50 a
ThryvOn Treated	28.00 ± 8.00 a	8.00 ± 8.00	48.00 ± 4.90 b	84.00 ± 9.80 b
ThryvOn Untreated	24.00 ± 4.00 a	8.00 ± 4.90	80.00 ± 14.14 b	112.00 ± 20.59 b
Trait *p value*	1.00	0.013	0.042	0.021
Insecticide *p value*	0.128	0.063	0.0001	0.001
Trait * Insecticide *p value*	0.048	0.063	0.022	0.001

Means in a column followed by a common letter are not significantly different (Tukey’s HSD, *p* < 0.05). * Denotes that there is only a significant trait effect. ** Denotes that there is only a significant spray effect. *** Denotes a significant interaction of insecticide treatments and trait effects.

**Table 2 toxins-15-00644-t002:** Percent (mean ± SEM) cotton fleahopper infested terminals by insect stage and week of squaring during 2020.

Treatment	First Week of Squaring			
	Small nymphs	Large nymphs	Adults *	Total *
Non-traited Treated	1.00 ± 0.58	0.00 ± 0.00	0.33 ± 0.33	1.33 ± 0.67
Non-traited Untreated	0.00 ± 0.00	0.00 ± 0.00	0.33 ± 0.33	0.33 ± 0.33
ThryvOn Treated	1.67 ± 0.33	0.00 ± 0.00	0.33 ± 1.20	5.00 ± 1.53
ThryvOn Untreated	2.00 ± 1.15	0.00 ± 0.00	3.00 ± 0.58	5.00 ± 1.73
Trait *p value*	0.081	1.000	0.004	0.009
Insecticide *p value*	0.631	1.000	0.820	0.691
Trait * Insecticide *p value*	0.347	1.000	0.820	0.691
	Second week of squaring			
	Small nymphs	Large nymphs	Adults *	Total *
Non-traited Treated	3.33 ± 1.2	0.00 ± 0.00	6.67 ± 2.03	10.00 ± 3.20
Non-traited Untreated	2.67 ± 0.67	0.67 ± 0.67	5.33 ± 1.2	8.67 ± 2.33
ThryvOn Treated	5.33 ± 1.67	1.33 ± 0.88	10.67 ± 1.33	17.33 ± 1.76
ThryvOn Untreated	6.33 ± 1.76	1.00 ± 0.58	12 ± 3.06	19.33 ± 4.91
Trait *p value*	0.0766	0.218	0.031	0.025
Insecticide *p value*	0.908	0.796	1.000	0.922
Trait * Insecticide *p value*	0.567	0.446	0.532	0.625
	Third week of squaring			
	Small nymphs	Large nymphs **	Adults **	Total **
Non-traited Treated	0.00 ± 0.00	0 ± 0	0.33 ± 0.33	0.33 ± 0.33
Non-traited Untreated	1.67 ± 0.33	0.67 ± 0.67	16 ± 4.04	18.33 ± 4.26
ThryvOn Treated	0.33 ± 0.33	0.00 ± 0.00	0.00 ± 0.00	0.33 ± 0.33
ThryvOn Untreated	6.00 ± 5.51	3.00 ± 1.00	14.33 ± 4.67	23.33 ± 8.82
Trait *p value*	0.423	0.088	0.755	0.624
Insecticide *p value*	0.221	0.016	0.001	0.003
Trait * Insecticide *p value*	0.490	0.088	0.834	0.624
	Fourth week of squaring			
	Small nymphs **	Large nymphs **	Adults ***	Total **
Non-traited Treated	0.33 ± 0.33	0.00 ± 0.00	7.00 ± 1.50 b	7.33 ± 1.45
Non-traited Untreated	15.67 ± 2.33	5.67 ± 1.45	19.33 ± 2.60 a	40.67 ± 4.81
ThryvOn Treated	0.33 ± 0.33	0.00 ± 0.00	6.33 ± 2.33 b	6.67 ± 2.40
ThryvOn Untreated	12.33 ± 2.4	7.00 ± 0.00	8.67 ± 1.45 b	28.00 ± 3.79
Trait *p value*	0.353	0.386	0.024	0.083
Insecticide *p value*	<0.0001	0.0001	0.007	<0.0001
Trait * Insecticide *p value*	0.353	0.386	0.040	0.113
	Fifth week of squaring			
	Small nymphs **	Large nymphs **	Adults	Total **
Non-traited Treated	1.33 ± 0.88	0.67 ± 0.33	4.33 ± 1.2	6.33 ± 1.45
Non-traited Untreated	17.00 ± 3.79	11.67 ± 1.76	5.00 ± 1.53	33.67 ± 4.84
ThryvOn Treated	2.00 ± 1.15	0.33 ± 0.33	2.67 ± 1.45	5.00 ± 2.64
ThryvOn Untreated	18.67 ± 4.09	8.67 ± 2.85	5.33 ± 0.88	32.67 ± 4.81
Trait *p value*	0.696	0.353	0.620	0.763
Insecticide *p value*	0.001	0.0004	0.233	<0.0001
Trait * Insecticide *p value*	0.866	0.453	0.491	0.966

Means in a column followed by a common letter are not significantly different (Tukey’s HSD, *p* < 0.05). * Denotes that there is only a significant trait effect. ** Denotes that there is only a significant spray effect. *** Denotes a significant interaction of insecticide treatments and trait effects.

**Table 3 toxins-15-00644-t003:** Percent (mean ± SEM) cotton fleahopper infested terminals by insect stage and week of squaring during 2021.

Treatment	First Week of Squaring			
	Small nymphs *	Large nymphs	Adults *	Total *
Non-traited Treated	1.13 ± 0.61	0.00 ± 0.00	1.63 ± 0.38	2.75 ± 0.88
Non-traited Untreated	0.25 ± 0.16	0.00 ± 0.00	0.75 ± 0.31	1.00 ± 0.38
ThryvOn Treated	1.75 ± 0.31	0.00 ± 0.00	2.38 ± 0.50	4.12 ± 0.74
ThryvOn Untreated	2.25 ± 0.84	0.00 ± 0.00	2.75 ± 0.59	5.00 ± 1.22
Trait *p value*	0.024	1.000	0.005	0.004
Insecticide *p value*	0.575	1.000	0.589	0.616
Trait * Insecticide *p value*	0.317	1.000	0.182	0.139
	Second week of squaring			
	Small nymphs **	Large nymphs **	Adults **	Total **
Non-traited Treated	0.00 ± 0.00	0.00 ± 0.00	0.13 ± 0.38	0.13 ± 0.13
Non-traited Untreated	0.75 ± 0.37	0.75 ± 0.37	1.75 ± 0.31	3.25 ± 0.86
ThryvOn Treated	0.13 ± 0.13	0.13 ± 0.13	0.50 ± 0.50	0.75 ± 0.37
ThryvOn Untreated	1.75 ± 0.49	0.88 ± 0.3	1.38 ± 0.59	4.00 ± 0.57
Trait *p value*	0.087	0.611	1.000	0.222
Insecticide *p value*	0.001	0.005	0.002	0.001
Trait * Insecticide *p value*	0.418	1.000	0.300	0.911
	Third week of squaring			
	Small nymphs **	Large nymphs **	Adults	Total **
Non-traited Treated	0.13 ± 0.13	0.88 ± 0.23	1.00 ± 0.33	2.00 ± 0.5
Non-traited Untreated	2.00 ± 0.57	2.13 ± 0.69	2.00 ± 0.33	6.13 ± 0.79
ThryvOn Treated	0.50 ± 0.27	0.50 ± 0.13	1.38 ± 0.46	2.00 ± 0.57
ThryvOn Untreated	3.50 ± 0.98	0.88 ± 0.49	1.13 ± 0.44	5.88 ± 0.83
Trait *p value*	0.121	0.078	0.532	0.881
Insecticide *p value*	0.0003	0.012	0.349	0.001
Trait * Insecticide *p value*	0.345	0.889	0.124	0.183
	Fourth week of squaring			
	Small nymphs ***	Large nymphs ***	Adults ***	Total ***
Non-traited Treated	0.88 ± 0.35 b	0.38 ± 0.18 b	0.88 ± 0.35 b	2.13 ± 0.61 bc
Non-traited Untreated	4.75 ± 1.05 a	4.63 ± 0.92 a	3.13 ± 0.64 a	12.50 ± 2.02 a
ThryvOn Treated	0.25 ± 0.16 b	0.00 ± 0.00 b	0.13 ± 0.13 b	0.38 ± 0.26 c
ThryvOn Untreated	2.50 ± 0.63 ab	1.50 ± 0.38 b	1.13 ± 0.3 b	5.13 ± 1.04 b
Trait *p value*	0.049	0.003	0.003	0.002
Insecticide *p value*	0.003	0.0001	0.003	0.0001
Trait * Insecticide *p value*	0.034	0.005	0.044	0.004

Means in a column followed by a common letter are not significantly different (Tukey’s HSD, *p* < 0.05). * Denotes that there is only a significant trait effect. ** Denotes that there is only a significant spray effect. *** Denotes a significant interaction of insecticide treatments and trait effects.

**Table 4 toxins-15-00644-t004:** Percent fruit retention (mean ± SEM) during the 4th week of squaring by year.

	2019 * **	2020 ***	2021 ***
Non-traited Treated	61.26 ± 2.98	86.39 ± 2.59 a	91.81 ± 0.52 a
Non-traited Untreated	30.53 ± 3.71	69.08 ± 3.81 b	87.63 ± 0.64 b
ThryvOn Treated	81.5 ± 3.62	82.61 ± 0.14 ab	90.77 ± 0.88 a
ThryvOn Untreated	51.70 ± 3.70	80.75 ± 4.72 ab	90.05 ± 0.40 a
Trait effect *p value*	<0.001	0.266	0.215
Insecticide effect *p value*	<0.001	0.020	0.002
Trait * Insecticide effect *p value*	0.898	0.047	0.013

Means in a column followed by a common letter are not significantly different (Tukey’s HSD, *p <* 0.05). * Denotes that there is only a significant trait effect. ** Denotes that there is only a significant insecticide spray effect. *** Denotes a significant interaction of spray and trait effects.

## Data Availability

Not applicable.

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
