# Peer review of "Field Evaluation of Cotton Expressing Mpp51Aa2 as a Management Tool for Cotton Fleahoppers, Pseudatomoscelis seriatus (Reuter)"

_toxins, 2023, doi:10.3390/toxins15110644_

Round 1

Reviewer 1 Report

Comments and Suggestions for Authors

Summary

The authors of the present study were the effectiveness of Mpp51 transgenic cotton (ThryvOn) alone and in combination with chemical spraying versus non traited cotton in controlling populations of Cotton fleahoppers is evaluated in a three year study.

Main findings of the study

Bt-cotton expressing Mpp51 is efficient for the control of  Pseudatomoscelis seriatus

In general the results offer a good perspective of the efficacy of the Bt-cooton for the control of fleahoppers overtime

Below are some suggestions for the authors,

Introduction

Line 66 - the Cry acts at the level of epithelial cells of the midgut and generate pores on their surface. The pores are not generated in the peritrophic membrane.

Results

2.1. Cotton fleahoppers populations

Here I miss a bit of explanation on what is evaluated and what is the hypothesis. The tables show that ThryvOn is combined with a sprayed treatment but there is no description on what chemicals or insecticides are sprayed.

Discussion

The discussion is well done, comparing the results with recent publications on the field and speculation on some of the unexpected results is well reasoned.

Reviewer 2 Report

Comments and Suggestions for Authors The MS shows data on the effectiveness of a Cry51Aa variant (Cry51Aa2.834_16, named as ThryvOn) to control cotton fleahoppers. The results show a trendency of lower infestation of nymphs or small nymphs when crops are treated with ThryvOn. I recommended the acceptance of the manuscript  

Reviewer 3 Report

Comments and Suggestions for Authors

This manuscript evaluated the field efficacy of cotton expressing Bt toxin genes against cotton fleahoppers. Results showed that the Bt-expressing plant did not give more protection than the non-traited plant unless insecticide was applied. Following points should be clarified.

1) The cotton line ThryvOn used in this study contains many Bt-toxin genes including Cry1Ac, Cry2Ab, Vip3Aa and Cry51Aa. Therefore, it is not clear if the effect observed in this study was from Cry51Aa alone or a combination effect with other toxins. It would be better if a cotton expressing only Cry51Aa was tested.

2) Insect infestation in the year 2021 (Table 3) was very low in all treatments. It is too low to give any meaning. 

3) Conclusion should be revised since the plant used in this study produced many Bt toxins and not only Cry51Aa. Insect control efficacy might come from a combination of many toxins.

Round 2

Reviewer 3 Report

Comments and Suggestions for Authors

All comments have been addressed in the revised manuscript.